# Learning Stochastic Shortest Path with Linear Function Approximation

## Abstract

We study the stochastic shortest path (SSP) problem in reinforcement learning with linear function approximation, where the transition kernel is represented as a linear mixture of unknown models. We call this class of SSP problems as linear mixture SSP. We propose a novel algorithm for learning the linear mixture SSP, which can attain a $\widetilde{\mathcal{O}}(dB_\star^{1.5}\sqrt{K/c_{\min}})$ regret. Here $K$ is the number of episodes, $d$ is the dimension of the feature mapping in the mixture model, $B_\star$ bounds the expected cumulative cost of the optimal policy, and $c_{\min} > 0$ is the lower bound of the cost function. Our algorithm also applies to the case when $c_{\min} = 0$, where a $\widetilde{\mathcal{O}}(K^{2/3})$ regret is guaranteed. To the best of our knowledge, this is the first algorithm with a sublinear regret guarantee for learning linear mixture SSP. In complement to the regret upper bounds, we also prove a lower bound of $\Omega(dB_\star\sqrt{K})$, which nearly matches our upper bound.

## 1 Introduction

The Stochastic Shortest Path (SSP) model refers to a type of reinforcement learning (RL) problems where an agent repeatedly interacts with a stochastic environment and aims to reach some specific goal state while minimizing the cumulative cost. Compared with other popular RL settings such as episodic and infinite-horizon Markov Decision Processes (MDPs), the horizon length in SSP is random, varies across different policies, and can potentially be infinite because the interaction only stops when arriving at the goal state. Therefore, the SSP model includes both episodic and infinite-horizon MDPs as special cases, and is comparably more general and of broader applicability. In particular, many goal-oriented real-world problems fit better into the SSP model, such as navigation and GO game (Andrychowicz et al., 2017; Nasiriany et al., 2019).

In recent years, there emerges a line of works on developing efficient algorithms and the corresponding analyses for learning SSP. Most of them consider the episodic setting, where the interaction between the agent and the environment proceeds in $K$ episodes (Cohen et al., 2020; Tarbouriech et al., 2020a). For tabular SSP models where the sizes of the action and state space are finite, Cohen et al. (2021) developed a finite-horizon reduction algorithm that achieves the minimax regret $\widetilde{\mathcal{O}}(B_\star\sqrt{SAK})$, where $B_\star$ is the largest expected cost of the optimal policy starting from any state, $S$ is the number of states and $A$ is the number of actions. In a similar setting, Tarbouriech et al. (2021b) proposed the first algorithm that is minimax optimal, parameter-free and horizon-free at the same time. However, the algorithms mentioned above only apply to tabular SSP problems where the state and action space are small. In order to deal with SSP problems with large state and action spaces, function approximation techniques (Yang & Wang, 2019; Jin et al., 2020; Jia et al., 2020; Zhou et al., 2021b; Wang et al., 2020b;a) are needed.

Following the recent line of work on model-based reinforcement learning with linear function approximation (Modi et al., 2020; Jia et al., 2020; Ayoub et al., 2020; Zhou et al., 2021b), we consider a linear mixture SSP model, which extends the tabular SSP. More specifically, we assume that the transition probability is parametrized by $\mathbb{P}(s'|s,a) = \langle\phi(s'|s,a),\boldsymbol{\theta}^*\rangle$ for all triplet $(s,a,s') \in \mathcal{S} \times \mathcal{A} \times \mathcal{S}$, where $\mathcal{S}$ is the state space and $\mathcal{A}$ is the action space. Here we assume that $\phi \in \mathbb{R}^d$ is a known ternary feature mapping, and $\boldsymbol{\theta}^* \in \mathbb{R}^d$ is an *unknown* model parameter vector that needs to be learned. Such a setting has been previously studied for episodic MDPs (Modi et al., 2020; Jia et al., 2020; Ayoub et al., 2020; Cai et al., 2020) and infinite-horizon discounted MDPs (Zhou et al., 2021b). Nevertheless, algorithms developed in these works do not apply to SSP since the horizon length is random as mentioned above.

To tackle the challenge of varying horizon length, we propose a model-based optimistic algorithm with linear function approximation, dubbed `LEVIS`, for learning the linear mixture SSP. At the core of our algorithm are a confidence set of the model parameters and a specially designed Extended Value Iteration (`EVI`) subroutine for computing the optimistic estimate of the value function, which together guarantee that the algorithm will reach the goal state in every episode. Compared with the EVI subroutine developed for infinite-horizon discounted MDPs (Zhou et al., 2021b), we introduce a shrinking factor $q \approx 1/t$ in our `EVI` with $t$ being the cumulative number of time steps, which guarantees the convergence of `EVI`. To compensate for the bias introduced by this shrinking factor, our algorithm performs lazy policy update, which is triggered by the doubling of the time interval between two policy updates or the doubling of the determinant of the covariance matrix. With all these algorithmic designs, our algorithm is guaranteed to achieve a $\widetilde{O}(dB_\star^{1.5}\sqrt{K/c_{\min}})$ regret when $c_{\min} > 0$. To the best of our knowledge, this is the first algorithm that enjoys a sublinear regret for linear mixture SSP.

It is worth noting that a recent work by Vial et al. (2021) studied a different linear SSP model that is similar to linear MDPs (Yang & Wang, 2019; Jin et al., 2020), where both the underlying transition probability and cost function are linear in a known $d$-dimensional feature mapping $\boldsymbol{\psi} \in \mathbb{R}^d$, i.e., $\mathbb{P}(s'|s,a) = \langle \boldsymbol{\psi}(s,a), \boldsymbol{\mu}(s) \rangle$ and $c(s,a) = \langle \boldsymbol{\psi}(s,a), \boldsymbol{\theta} \rangle$, and $\boldsymbol{\mu}(\cdot)$ and $\boldsymbol{\theta}$ are unknown. They proposed an algorithm with linear function approximation, which achieves $\widetilde{\mathcal{O}}(\sqrt{B_\star^3 d^3 K/c_{\min}})$ regret. The linear SSP model is different from our model, and we refer the interested readers to Ayoub et al. (2020); Zhou et al. (2021b) for a detailed comparison between these two assumptions. Besides the model difference, Vial et al. (2021) further assumed the feature mapping to be orthonormal in order to obtain the $\widetilde{O}(\sqrt{K})$ regret. We do not need such restrictive assumptions on the feature mapping, thus our algorithm provably works for more general cases.

Our contributions are summarized as follows:

- We propose to study a linear mixture SSP model, and devise a novel algorithm, dubbed **L**ower confidence **E**xtended **V**alue **I**teration for **S**SP (`LEVIS`), for learning SSP with linear function approximation.
- We prove that `LEVIS` achieves a regret of order $\widetilde{\mathcal{O}}(B_\star^{1.5} d\sqrt{K/c_{\min}})$ when $c_{\min} > 0$ and the agent has an order-accurate estimate $B \geq B_\star$[1]. For the general case where $c_{\min} = 0$, our algorithm can achieve $\widetilde{\mathcal{O}}(K^{2/3})$ regret guarantee by using a cost perturbation trick (Tarbouriech et al., 2021b).
- We prove that for linear mixture SSP, the regret of any learning algorithms is at least $\Omega(dB_\star\sqrt{K})$. This suggests that when $c_{\min} > 0$, our algorithm is optimal with regard to the dimension of the feature mapping $d$ and number of episodes $K$.

**Notation** We use lower case letters to denote scalars, and use lower and upper case bold face letters to denote vectors and matrices respectively. For any positive integer $n$, we denote by $[n]$ the set $\{1, \ldots, n\}$. For a vector $\mathbf{x} \in \mathbb{R}^d$, we denote by $\|\mathbf{x}\|_1$ the Manhattan norm and denote by $\|\mathbf{x}\|_2$ the Euclidean norm. For a vector $\mathbf{x} \in \mathbb{R}^d$ and matrix $\boldsymbol{\Sigma} \in \mathbb{R}^{d \times d}$, we define $\|\mathbf{x}\|_{\boldsymbol{\Sigma}} = \sqrt{\mathbf{x}^\top \boldsymbol{\Sigma} \mathbf{x}}$. For two sequences $\{a_n\}$ and $\{b_n\}$, we write $a_n = O(b_n)$ if there exists an absolute constant $C$ such that $a_n \leq Cb_n$. We use $\widetilde{O}(\cdot)$ to hide the logarithmic factors.

## 2 RELATED WORK

**Online learning in SSP** SSP problems can be dated back to (Bertsekas & Tsitsiklis, 1991; Bertsekas & Yu, 2013; Bertsekas, 2012), but it is until recently that the regret minimization in online learning of SSP has been studied. In the tabular case, Tarbouriech et al. (2020a) proposed the first algorithm achieving a $\widetilde{\mathcal{O}}(D^{3/2}S\sqrt{AK/c_{\min}})$ regret where $D$ is the diameter of SSP[2]. The regret was further improved to $\widetilde{O}(B_\star S\sqrt{AK})$ by Rosenberg et al. (2020); Cohen et al. (2020), with an extra $\sqrt{S}$ factor compared with the $\Omega(B_\star\sqrt{SAK})$ lower bound (Rosenberg et al., 2020). More recently, the $\widetilde{O}(B_\star\sqrt{SAK})$ minimax optimal regret were obtained by Cohen et al. (2021) and Tarbouriech et al. (2020b) independently using different approaches. Specifically, Cohen et al. (2021) reduced SSP to a finite-horizon MDP with a large terminal cost assuming $B_\star$ is known; while Tarbouriech et al. (2021b) avoid such requirement by adaptively estimating $B_\star$ with a doubling trick, together with

---

[1]We say $B$ is an order-accurate estimate of $B^*$, if there exists some unknown constant $\kappa \geq 1$ such that $B_\star \leq B \leq \kappa B_\star$.

[2]The diameter of an SSP is defined as the longest possible shortest path from any initial state to the goal state.

a value iteration sub-routine ensuring the optimistic estimate of the value function. Our proposed method shares a similar spirit with the latter approach, but for learning SSP with linear function approximation.

The above algorithms are all model-based. Very recently, Chen et al. (2021a) developed the first model-free algorithm for SSP which achieves the minimax optimal regret when the minimum cost among all state-action pairs $c_{\min}$ is strictly positive. Their method is motivated by the `UCB-ADVANTAGE` algorithm (Zhang et al., 2020). For other settings of SSP, (Rosenberg & Mansour, 2020; Chen & Luo, 2021; Chen et al., 2021b) studied the case of adversarial costs. Also, the pioneering work by (Bertsekas & Tsitsiklis, 1991) studied the pure planning problem in SSP where the agent has full knowledge of all the model parameters, and is followed by a series of works (Bonet, 2007; Kolobov et al., 2011; Bertsekas & Yu, 2013; Guillot & Stauffer, 2020). On the other hand, Tarbouriech et al. (2021a) studied the sample complexity of SSP assuming the access to a generative model. Jafarnia-Jahromi et al. (2021) proposed the first posterior sampling algorithm for SSP. Multi-goal SSP have also been studied by Lim & Auer (2012); Tarbouriech et al. (2020b).

**Linear function approximation** Linear MDP is one of the most widely studied models for RL with linear function approximation, which assumes both the transition probability and reward functions are linear functions of a known feature mapping (Yang & Wang, 2019; Jin et al., 2020). Representative work in this direction include Du et al. (2019); Zanette et al. (2020); Wang et al. (2020a); He et al. (2021), to mention a few.

Another popular model for RL with linear function approximation is the so-called linear mixture MDP/linear kernel MDP (Yang & Wang, 2020; Modi et al., 2020; Jia et al., 2020; Ayoub et al., 2020; Cai et al., 2020; Zhou et al., 2021b;a). For the finite-horizon setting, Jia et al. (2020) proposed a `UCRL-VTR` algorithm that achieves a $\widetilde{\mathcal{O}}(d\sqrt{H^3 T})$ regret bound. Zhou et al. (2021a) further improve the result by proposing a `UCRL-VTR+` algorithm that attains the nearly minimax optimal regret $\widetilde{\mathcal{O}}(dH\sqrt{T})$ based on a novel Bernstein-type concentration inequality. For the discounted infinite horizon setting, Zhou et al. (2021b) proposed a `UCLK` algorithm with a $\widetilde{\mathcal{O}}(d\sqrt{T}/(1-\gamma)^2)$ regret, and also give a $\widetilde{\mathcal{O}}(d\sqrt{T}/(1-\gamma)^{1.5})$ lower bound. The lower bound is later matched up to logarithmic factors by the `UCLK+` algorithm (Zhou et al., 2021a). The SSP model studied in this paper can be seen as an extension of linear mixture MDPs.

## 3 PRELIMINARIES

**Stochastic Shortest Path** An SSP instance is an MDP $M := \{\mathcal{S}, \mathcal{A}, \mathbb{P}, c, s_{\mathrm{init}}, g\}$, where $\mathcal{S}$ and $\mathcal{A}$ are the finite state space and action space respectively. Here $s_{\mathrm{init}}$ denotes the initial state and $g \in \mathcal{S}$ is the goal state. We denote the cost function by $c : \mathcal{S} \times \mathcal{A} \to [0, 1]$, where $c(s, a)$ is the immediate cost of taking action $a$ at state $s$. The goal state $g$ incurs zero cost, i.e., $c(g, a) = 0$ for all $a \in \mathcal{A}$. For any $(s', s, a) \in \mathcal{S} \times \mathcal{A} \times \mathcal{S}$, $\mathbb{P}(s'|s, a)$ is the probability to transition to $s'$ given the current state $s$ and action $a$ being taken. The goal state $g$ is an absorbing state, i.e., $\mathbb{P}(g|g, a) = 1$ for all action $a \in \mathcal{A}$.

**Linear mixture SSP** In this work, we assume the transition probability function $\mathbb{P}$ to be a linear mixture of some basis kernels (Modi et al., 2020; Ayoub et al., 2020; Zhou et al., 2021a).

**Assumption 3.1.** Suppose the feature mapping $\phi : \mathcal{S} \times \mathcal{A} \times \mathcal{S} \to \mathbb{R}^d$ is known and pregiven. There exists an *unknown* vector $\boldsymbol{\theta}^* \in \mathbb{R}^d$ with $\|\boldsymbol{\theta}^*\|_2 \le \sqrt{d}$ such that $\mathbb{P}(s'|s, a) = \langle \phi(s'|s, a), \boldsymbol{\theta}^* \rangle$ for any state-action-state triplet $(s, a, s') \in \mathcal{S} \times \mathcal{A} \times \mathcal{S}$. Moreover, for any bounded function $V : \mathcal{S} \to [0, B]$, it holds that $\|\phi_V(s, a)\|_2 \le B\sqrt{d}$ for all $(s, a) \in \mathcal{S} \times \mathcal{A}$, where $\phi_V(s, a) := \sum_{s' \in \mathcal{S}} \phi(s'|s, a)V(s')$.

For simplicity, for any function $V : \mathcal{S} \to \mathbb{R}$, we denote $\mathbb{P}V(s, a) = \sum_{s'} \mathbb{P}(s'|s, a)V(s')$ for all $(s, a) \in \mathcal{S} \times \mathcal{A}$. Therefore, under Assumption 3.1, we have

$$\mathbb{P}V(s, a) = \sum_{s' \in \mathcal{S}} \mathbb{P}(s'|s, a)V(s') = \sum_{s' \in \mathcal{S}} \langle \phi(s'|s, a), \boldsymbol{\theta}^* \rangle V(s') = \langle \phi_V(s, a), \boldsymbol{\theta}^* \rangle.$$

**Proper policies** A stationary and deterministic policy is a mapping $\pi : \mathcal{S} \to \mathcal{A}$ such that the action $\pi(s)$ is taken given the current state $s$. We denote by $T^\pi(s)$ the expected time that it takes by following $\pi$ to reach the goal state $g$ starting from $s$. We say a policy $\pi$ is proper if $T^\pi(s) < \infty$ for any $s \in \mathcal{S}$ (otherwise it is improper). We denote by $\Pi_{\mathrm{proper}}$ the set of all stationary, deterministic and proper policies. We assume that $\Pi_{\mathrm{proper}}$ is non-empty, which is the common assumption in previous works on online learning of SSP (Rosenberg et al., 2020; Rosenberg & Mansour, 2020; Cohen et al., 2021; Tarbouriech et al., 2021b; Jafarnia-Jahromi et al., 2021; Chen et al., 2021a).

**Assumption 3.2.** The set of all stationary, deterministic and proper policies is non-empty, i.e., $\Pi_{\text{proper}} \neq \varnothing$.

**Remark 3.3.** The above assumption is weaker than Assumption 1 in Vial et al. (2021) which requires that all stationary policies are proper.

For any policy $\pi$, we define the cost-to-go function (a.k.a., value function) as

$$V^\pi(s) := \lim_{T \to +\infty} \mathbb{E}\left[\sum_{t=1}^{T} c(s_t, \pi(s_t)) \,\middle|\, s_1 = s\right], \quad \text{where } s_{t+1} \sim \mathbb{P}\big(\,\cdot\,|s_t, \pi(s_t)\big).$$

$V^\pi(s)$ can possibly be infinite if $\pi$ is improper. The action-value function of policy $\pi$ is defined as

$$Q^\pi(s, a) := \lim_{T \to \infty} \mathbb{E}\left[c(s_1, a_1) + \sum_{t=2}^{T} c(s_t, \pi(s_t)) \,\middle|\, s_1 = s,\ a_1 = a\right],$$

where $s_2 \sim \mathbb{P}(\cdot|s_1, a_1)$ and $s_{t+1} \sim \mathbb{P}(\cdot|s_t, \pi(s_t))$ for all $t \geq 2$. Since $c(\cdot, \cdot) \in [0, 1]$, for any proper policy $\pi \in \Pi_{\text{proper}}$, $V^\pi$ and $Q^\pi$ are both bounded functions.

**Bellman optimality** For any function $V : \mathcal{S} \to \mathbb{R}$, we define the optimal Bellman operator $\mathcal{L}$ as

$$\mathcal{L}V(s) := \min_{a \in \mathcal{A}}\{c(s, a) + \mathbb{P}V(s, a)\}. \tag{3.1}$$

Intuitively speaking, we want to learn the optimal policy $\pi^\star$ such that $V^\star(\cdot) := V^{\pi^\star}(\cdot)$ is the unique solution to the Bellman optimality equation $V = \mathcal{L}V$ and $\pi^\star$ minimizes the value function $V^\pi(s)$ component-wise over all policies. It is known that, in order for such $\pi^\star$ to exist, one sufficient condition is Assumption 3.2 together with an extra condition that any improper policy $\pi$ has at least one infinite-value state, i.e., for any $\pi \notin \Pi_{\text{proper}}$, there exists some $s \in \mathcal{S}$ s.t. $V^\pi(s) = +\infty$ (Bertsekas & Tsitsiklis, 1991; Bertsekas & Yu, 2013; Tarbouriech et al., 2021b). Note that this additional condition is satisfied in the case of strictly positive cost , where for any state $s \neq g$ and $a \in \mathcal{A}$, it holds that $c(s, a) \geq c_{\min}$. To deal with the case of general cost function, one can adopt the cost perturbation trick (Tarbouriech et al., 2021b) and consider a modified problem with cost function $c_\rho(s, a) := \max\{c(s, a), \rho\}$ for some $\rho > 0$. This will introduce an additional cost of order $\mathcal{O}(\rho T)$ to the regret of the original problem, where $T$ is the total number of steps. Therefore, the second condition can be avoided, and we can assume the existence of $\pi^\star$.

Throughout the paper, we denote by $B_\star$ the upper bound of the optimal value function $V^\star$, i.e., $B_\star := \max_{s \in \mathcal{S}} V^\star(s)$. Also, we define $T_\star := \max_{s \in \mathcal{S}} T^{\pi^\star}(s)$, which is finite under Assumption 3.2. Since the cost is bounded by 1, we have $B_\star \leq T_\star < +\infty$. Without loss of generality, we assume that $B_\star \geq 1$. Furthermore, we denote the corresponding optimal action-value function by $Q^\star := Q^{\pi^\star}$ which satisfies the following Bellman equation for all $(s, a) \in \mathcal{S} \times \mathcal{A}$:

$$Q^\star(s, a) = c(s, a) + \mathbb{P}V^\star(s, a), \quad V^\star(s) = \min_{a \in \mathcal{A}} Q^\star(s, a). \tag{3.2}$$

**Learning objective** Under Assumption 3.1, we assume $c$ to be known for the ease of presentation. We study the episodic setting where each episode starts from a fixed initial state $s_{\text{init}}$ and ends only if the agent reaches the goal state $g$. Given the total number of episodes, $K$, the objective of the agent is to minimize the regret over $K$ episodes defined as

$$R_K := \sum_{k=1}^{K} \sum_{i=1}^{I_k} c_{k,i} - K \cdot V^\star(s_{\text{init}}), \tag{3.3}$$

where $I_k$ is the length of the $k$-th episode and $c_{k,i} = c(s_{k,i}, a_{k,i})$ is the cost triggered at the $i$-th step during the $k$-th episode. Note that $R_K$ might be infinite if some episode never ends.

## 4 ALGORITHMS

In this section, we propose a model-based algorithm named `LEVIS`, as displayed in Algorithm 1. `LEVIS` is inspired by the `UCLK`-type of algorithms originally designed for discounted linear mixture MDPs (Zhou et al., 2021a;b). Our algorithm takes a multi-epoch form, where each episode is divided into epochs of different lengths (Jaksch et al., 2010; Lattimore & Hutter, 2012). Within each epoch, the agent executes the greedy policy induced by some optimistic estimator of the optimal Q-function. The switch between any two epochs is triggered by a doubling criterion, and then the estimated Q-function is updated through an Extend Value Iteration (`EVI`) sub-routine (Algorithm 2). We now give a detailed description of Algorithm 1.

---

**Algorithm 1** `LEVIS`

---

1: **Input:** regularization parameter $\lambda$, confidence radius $\{\beta_t\}$, cost perturbation $\rho \in [0,1]$, an estimate $B \geq B_\star$
2: **Initialize:** set $t \leftarrow 1$, $j \leftarrow 0$, $t_0 = 0$, $\boldsymbol{\Sigma}_0 \leftarrow \lambda \mathbf{I}$, $\mathbf{b}_0 \leftarrow \mathbf{0}$, $Q_0(s,\cdot), V_0(s) \leftarrow 1 \; \forall s \neq g$ and 0 otherwise
3: **for** $k = 1, \ldots, K$ **do**
4:    Set $s_t = s_{\text{init}}$
5:    **while** $s_t \neq g$ **do**
6:       Take action $a_t = \operatorname{argmin}_{a \in \mathcal{A}} Q_j(s_t, a)$, receive cost $c_t = c(s_t, a_t)$ and next state $s_{t+1} \sim \mathbb{P}(\cdot | s_t, a_t)$
7:       Set $\boldsymbol{\Sigma}_t \leftarrow \boldsymbol{\Sigma}_{t-1} + \boldsymbol{\phi}_{V_j}(s_t, a_t) \boldsymbol{\phi}_{V_j}(s_t, a_t)^\top$
8:       Set $\mathbf{b}_t \leftarrow \mathbf{b}_{t-1} + \boldsymbol{\phi}_{V_j}(s_t, a_t) V_j(s_{t+1})$
9:       **if** $\det(\boldsymbol{\Sigma}_t) \geq 2 \det(\boldsymbol{\Sigma}_{t_j})$ **or** $t \geq 2t_j$ **then**
10:          Set $j \leftarrow j + 1$
11:          Set $t_j \leftarrow t$, $\epsilon_j \leftarrow \frac{1}{t_j}$
12:          $\widehat{\boldsymbol{\theta}}_j \leftarrow \boldsymbol{\Sigma}_t^{-1} \mathbf{b}_t$
13:          Set confidence set $\mathcal{C}_j \leftarrow \left\{ \boldsymbol{\theta} : \|\boldsymbol{\Sigma}_{t_j}^{1/2}(\boldsymbol{\theta} - \widehat{\boldsymbol{\theta}}_j)\|_2 \leq \beta_{t_j} \right\}$
14:          Set $Q_j(\cdot,\cdot) \leftarrow \texttt{EVI}(\mathcal{C}_j, \epsilon_j, \frac{1}{t_j}, \rho)$
15:          Set $V_j(\cdot) \leftarrow \min_{a \in \mathcal{A}} Q_j(\cdot, a)$
16:       **end if**
17:       Set $t \leftarrow t + 1$
18:    **end while**
19: **end for**

---

**Algorithm 2** `EVI`

---

1: **Input:** confidence set $\mathcal{C}$, error parameter $\epsilon$, transition bonus $q$, cost perturbation $\rho \in [0,1]$
2: **Initialize:** $i \leftarrow 0$, and $Q^{(0)}(\cdot,\cdot)$, $V^{(0)}(\cdot) = 0$, and $V^{(-1)}(\cdot) = +\infty$
3: Set $Q(\cdot,\cdot) \leftarrow Q^{(0)}(\cdot,\cdot)$
4: **if** $\mathcal{C} \cap \mathcal{B} \neq \phi$ **then**
5:    **while** $\|V^{(i)} - V^{(i-1)}\|_\infty \geq \epsilon$ **do**
6:

$$Q^{(i+1)}(\cdot,\cdot) \leftarrow c_\rho(\cdot,\cdot) + (1-q) \cdot \min_{\boldsymbol{\theta} \in \mathcal{C} \cap \mathcal{B}} \langle \boldsymbol{\theta}, \boldsymbol{\phi}_{V^{(i)}}(\cdot,\cdot) \rangle \tag{4.1}$$

$$V^{(i+1)}(\cdot) \leftarrow \min_{a \in \mathcal{A}} Q^{(i+1)}(\cdot,a) \tag{4.2}$$

7:       Set $i \leftarrow i + 1$
8:    **end while**
9:    $Q(\cdot,\cdot) \leftarrow Q^{(i+1)}(\cdot,\cdot)$
10: **end if**
11: **Output:** $Q(\cdot,\cdot)$

---

In Algorithm 1, we maintain two global indices. Index $t$ represents the total number of steps, and index $j$ tracks the number of calls to the `EVI` sub-routine, where the output of `EVI` is an updated optimistic estimator of the optimal action-value function. Each episode starts from a fixed initial state $s_{\text{init}}$ (Line 4), ends when the goal state $g$ is reached (Line 5) and is decomposed into epochs indexed by the global index $j$. Within epoch $j$, the agent repeatedly executes the policy induced by the current estimation $Q_j$ of the action-value function (Line 6) and updates $\boldsymbol{\Sigma}_t$ and $\mathbf{b}_t$ (Lines 7 and 8). The current epoch ends when the either criterion in Line 9 is triggered, and the `EVI` subroutine performs an optimistic planning to update the action-value function estimator (Lines 10 to 15).

**Update criteria** As mentioned before, Algorithm 1 runs in epochs indexed by $j$, and one epoch ends when either of the two update criteria is triggered (Line 9). The first updating criterion is satisfied once the determinant of $\boldsymbol{\Sigma}_t$ is doubled compared to the determinant at the end of the previous epoch. This

is called lazy policy update that has been used in the linear bandits and RL literature (Abbasi-Yadkori et al., 2011; Zhou et al., 2021b), which reflects the diminishing return of learning the underlying transition. One intuition behind the determinant doubling criterion is that the determinant can be viewed as a surrogate measure of the exploration in the feature space. Thus, one only updates the policy when there is enough exploration being made since last update. Moreover, this update criterion reduces the computational cost as the total number of epochs would be bounded by $\mathcal{O}(\log T)$. Here $T$ denotes the total number of steps through all $K$ episodes. The doubling visitation criterion used in tabular SSP (Jafarnia-Jahromi et al., 2021; Tarbouriech et al., 2021b) can be viewed as a special case of this doubling determinant-based criterion.

However, the above criterion alone cannot guarantee finite length for each epoch as we do not have that $\|\phi_V(\cdot, \cdot)\|$ is bounded from below, which holds for tabular SSP naturally since at most $|\mathcal{S}||\mathcal{A}| \max_{s \in \mathcal{S}, a \in \mathcal{A}} n(s, a)$ steps suffice to double $n(s, a)$ for at least a pair of $s, a$ by the pigeonhole principle. To address this problem, we show that we only need to add an extra triggering criterion: $t \geq 2t_j$. It turns out that despite of being extremely simple this criterion endows the algorithm with several nice properties. First, together with the EVI error parameter $\epsilon_j = 1/t_j$, we can bound the cumulative error from value iterations in epoch $j$ by a constant, i.e., $(2t_j - t_j) \cdot \epsilon_j = 1$. Second, it will not increase the total number of epochs since the time step doubling can happen at most $\mathcal{O}(\log T)$ times, which is consistent with the first criterion. These two properties together allow us to bound the total error from value iteration by $\mathcal{O}(\log T)$. Finally, this criterion is fairly easy to implement and has negligible time and space complexity.

**Optimistic planning** The optimism of Algorithm 1 is realized by the construction of the confidence set $\mathcal{C}_j$ (Line 11), which is fed into the EVI subroutine. We now describe the construction of the Q-function estimator in the EVI sub-routine (Algorithm 2). EVI requires the access to a confidence ellipsoid $\mathcal{C}_j$ that contains the true model parameter $\boldsymbol{\theta}^*$ with high probability (Line 13). Here we construct the confidence set $\mathcal{C}_j$ centered at the minimizer of the ridge regression problem with a confidence radius parameter $\beta_t$ (Line 13). Since not every $\boldsymbol{\theta} \in \mathcal{C}_j$ defines a valid transition probability function, we further take the intersection between $\mathcal{C}_j$ and a constraint set $\mathcal{B}$ defined as follows

$$\mathcal{B} := \{\boldsymbol{\theta} : \forall (s, a), \langle \phi(\cdot|s, a), \boldsymbol{\theta} \rangle \text{ is a probability distribution and } \langle \phi(s'|g, a), \boldsymbol{\theta} \rangle = \mathbb{1}\{s' = g\}\}.$$

Then $\mathcal{C}_j \cap \mathcal{B}$ is still a confidence set containing the true model parameter $\boldsymbol{\theta}^*$ with high probability as $\boldsymbol{\theta}^* \in \mathcal{B}$. Algorithm 2 requires two additional inputs: optimality gap $\epsilon_j$ and discount factor $q$. The use of $\epsilon_j$ is standard, but this discount factor is the key to ensuring convergence of EVI.

Specifically, (4.1) in Algorithm 2 repeatedly conducts one-step value iteration by applying the best possible Bellman operator to the set $\mathcal{C}_j \cap \mathcal{B}$. This is motivated by the Bellman optimality equation in (3.2), and uses $\min_{\boldsymbol{\theta} \in \mathcal{C} \cap \mathcal{B}} \langle \boldsymbol{\theta}, \phi_{V^{(i)}} \rangle$ as an optimistic estimate for $\mathbb{P}V^*$. However, using this estimate alone cannot guarantee the convergence of EVI because $\langle \cdot, \phi_{V^{(i)}} \rangle$ is not a contractive map, which holds for free in the discounted setting (Jaksch et al., 2010; Zhou et al., 2021b), but not in SSP. More specifically, in the EVI algorithm for the discounted setting (e.g., Algorithm 2 in (Zhou et al., 2021b)), there is an intrinsic discount factor $0 < \gamma < 1$, which ensures that the Bellman operator is a contraction. As a result, the value iteration converges in a finite number of iterations. However, the Bellman equation of SSP does not have a discount factor. To address this issue, in (4.1), we introduce an extra $1 - q$ discount factor to ensure the contraction property. Although this causes an additional bias to the estimated transition probability function, we can alleviate it by choosing $q$ properly. In particular, for each epoch $j$ we set $q = 1/t_j$ (Line 14), and as will be shown, this bias will only introduce an additive term of order $\mathcal{O}(\log T)$ in the final regret bound.

Besides the convergence guarantee, the $1 - q$ factor also brings an additional benefit that it biases the estimated transition kernel towards the goal state $g$, further encouraging optimism. Similar design can also be found in the VISGO value iteration algorithm used by Tarbouriech et al. (2021b). The intuition behind such a design is to ensure the existence of proper policies under the estimated transition probability function. As a result, the output of the value iteration, which solves $V = \widetilde{\mathcal{L}}V$ approximately for the Bellman operator $\widetilde{\mathcal{L}}$ induced by the estimated transition, can induce a greedy policy that is proper under the estimated transition.

Regarding the implementation of LEVIS, note that the main computational overhead is from EVI, where within each inner iteration we need to solve an optimization problem. Fortunately, the loss function is strongly convex, thus it can be efficiently solved by many convex optimization algorithms.

## 5 Main Results

In this section, we present the main theoretical results for Algorithm 1. We provide regret upper bounds for both positive cost functions and general cost functions, followed by a lower bound.

### 5.1 Upper Bounds: Positive Cost Functions

We first consider a special case where the cost is strictly positive (except for the goal state $g$).

**Assumption 5.1.** There exists an *unknown* constant $c_{\min} \in (0, 1)$ such that $c(s, a) \geq c_{\min}$ for all $s \in \mathcal{S} \setminus \{g\}$ and $a \in \mathcal{A}$.

Let $T$ be the total number of steps in Algorithm 1, then the above assumption allows us to lower bound the total cumulative cost after the $K$ episodes by $c_{\min} \cdot T$. Note that this provides a relation between the deterministic $K$ and the random quantity $T$. To simplify the expression, we assume the agent has access to $B$, an order-accurate estimate of $B_\star$ satisfying $B_\star \leq B \leq \kappa B_\star$ for some unknown constant $\kappa \geq 1$. Similar assumptions have also been imposed in previous works (Tarbouriech et al., 2021b; Vial et al., 2021).

**Theorem 5.2.** Under Assumptions 3.1, 3.2 and 5.1, for any $\delta > 0$, let $\rho = 0$ and $\beta_t = B\sqrt{d \log\left(4(t^2 + t^3 B^2/\lambda)/\delta\right)} + \sqrt{\lambda d}$ for all $t \geq 1$, where $B \geq B_\star$ and $\lambda \geq 1$. Then with probability at least $1 - \delta$, the regret of Algorithm 1 satisfies

$$R_K = \mathcal{O}\left(B^{1.5}d\sqrt{K/c_{\min}} \cdot \log^2\left(\frac{KBd}{c_{\min}\delta}\right) + \frac{B^2 d^2}{c_{\min}}\log^2\left(\frac{KBd}{c_{\min}\delta}\right)\right). \tag{5.1}$$

If $B = O(B_\star)$, Algorithm 1 attains an $\widetilde{O}(B_\star^{1.5}d\sqrt{K/c_{\min}})$ regret. The dominating term in (5.1) has an dependency on $1/c_{\min}$. For the tabular SSP, Cohen et al. (2021); Jafarnia-Jahromi et al. (2021); Tarbouriech et al. (2021b) avoid such dependency by using a Bernstein-type confidence set. However, it remains an open question whether a similar result can be achieved under the linear function approximation setting.

**Remark 5.3.** If we set the parameter $\delta$ in Theorem 5.2 as $\delta = 1/K$ and define the high probability event $\Omega$ as Theorem 5.2 holds. Then, for the expected regret, we have

$$\mathbb{E}[R_K] \leq \mathbb{E}\left[R_K | \Omega\right] \Pr[\Omega] + K Pr[\bar{\Omega}]$$
$$= \mathcal{O}\left(B^{1.5}d\sqrt{K/c_{\min}} \cdot \log^2\left(\frac{KBd}{c_{\min}}\right) + \frac{B^2 d^2}{c_{\min}}\log^2\left(\frac{KBd}{c_{\min}}\right)\right),$$

which implies an $\widetilde{O}(B_\star^{1.5}d\sqrt{K/c_{\min}})$ expected regret.

### 5.2 Upper Bound: General Cost Functions

When Assumption 5.1 does not hold, an $\widetilde{O}(K^{2/3})$ regret can be achieved by running Algorithm 1 with $\rho = K^{-1/3}$.

**Theorem 5.4.** Under Assumptions 3.1 and 3.2, for any $\delta > 0$, let $\rho = K^{-1/3}$ and $\beta_t = B\sqrt{d \log\left(4(t^2 + t^3 B^2/\lambda)/\delta\right)} + \sqrt{\lambda d}$ for all $t \geq 1$, where $B \geq B_\star$ and $\lambda \geq 1$. Then with probability at least $1 - \delta$, the regret of Algorithm 1 satisfies

$$R_K = \mathcal{O}\left(\widetilde{B}^{1.5}dK^{2/3} \cdot \chi + T_\star K^{2/3} + \widetilde{B}^2 d^2 K^{1/3} \cdot \chi\right),$$

where $\widetilde{B} = B + T_\star/K^{1/3}$ and $\chi = \log^2\left((B + T_\star)Kd/\delta\right)$.

In Theorem 5.4, the regret depends on $\widetilde{B}$ instead of $B$. Note that $\widetilde{B}$ is approximately equal to $B_\star$ when $K = \Omega(T_\star^3)$ and $B = O(B_\star)$. Here $T_\star$ is defined in Section 3 as the maximum expected time it takes for the optimal policy to reach the goal state starting from any state.

The cost perturbation $\rho$ is a common trick to deal with the case of general cost functions in the SSP literature (Tarbouriech et al., 2020a; Cohen et al., 2020; Tarbouriech et al., 2021b). Similar to Tarbouriech et al. (2020a), the term $c_{\min}^{-1}$ is multiplicative with $K$ in our regret bound given by Theorem 5.2. As a result, the perturbation can only give an $\widetilde{\mathcal{O}}(K^{2/3})$ regret in the case of general cost functions. Similarly, the regret bound of learning linear SSP (Vial et al., 2021) also has a multiplicative $c_{\min}^{-1}$. Some later work on tabular SSP (Cohen et al., 2020; Tarbouriech et al., 2021b) has shown that it is possible to make the term $c_{\min}^{-1}$ additive instead of multiplication, which improves the regret to $\widetilde{\mathcal{O}}(K^{1/2})$ for general cost functions. How to get an additive $c_{\min}^{-1}$ term in the linear function approximation setting is an interesting future direction.

For the choice of the other parameters in Algorithm 1, by Theorems 5.2 and 5.4, we can set $\lambda = 1$ in both the positive and general cost cases. For the upper bound $B \geq B_\star$, note that assuming a known $B$ is common in existing SSP literature (Cohen et al., 2021; Vial et al., 2021). Although it is possible to deal with an unknown $B$ in the tabular SSP with a doubling trick (Rosenberg et al., 2020; Tarbouriech et al., 2021b), it remains an open question for SSP with linear function approximation.

### 5.3 Lower Bound

We also provide a hardness result for learning linear mixture SSP by proving the lower bound for the expected regret suffered by any deterministic learning algorithms.

**Theorem 5.5.** Under Assumption 3.1, suppose $d \geq 2$, $B_\star \geq 2$ and $K > (d-1)^2/2^{12}$. Then for any possibly non-stationary history-dependent policy $\pi$, there exists a linear mixture SSP instance with parameter $\boldsymbol{\theta}^*$ such that

$$\mathbb{E}_{\pi,\boldsymbol{\theta}^*}[R_K] \geq \frac{dB_\star\sqrt{K}}{1024}. \tag{5.2}$$

**Remark 5.6.** The expectation in (5.2) is over the trajectories induced by executing the policy $\pi$ in the SSP environment parameterized by $\boldsymbol{\theta}^*$. Note that here we allow the policy $\pi$ to be non-stationary and history-dependent. This is equivalent to assuming a deterministic learning algorithm, which is sufficient for establishing a lower bound (Cohen et al., 2020).

**Remark 5.7.** Our instance for the lower bound can be also adapted to a linear SSP instance (Vial et al., 2021), which yields a $\Omega(dB_\star\sqrt{K})$ lower bound. (See Remark E.1 for a detailed discussion.)

## 6 Proof Sketch of the Main Results

In this section, we give a proof sketch of the main results in Section 5. Due to space limit, we defer the proof of the lemmas to the appendix.

### 6.1 Proof of Theorem 5.2

In this subsection, we prove Theorem 5.2, which gives the regret upper bound of Algorithm 1 for positive cost functions. The proof relies on the following intermediate result.

**Theorem 6.1.** Under Assumption 3.1 and 3.2, for any $\delta > 0$, let $\rho = 0$ and $\beta_t = B\sqrt{d\log\left(4(t^2 + t^3B^2/\lambda)/\delta\right)} + \sqrt{\lambda d}$ for some $B \geq B_\star$ where $\lambda \geq 1$ and $\rho = 0$. Then with probability at least $1 - \delta$, the regret of Algorithm 1 satisfies

$$R_K \leq 6\beta_T\sqrt{dT\log\left(1 + \frac{TB_\star^2}{\lambda}\right)} + 7dB_\star\log\left(T + \frac{T^2B_\star^2d}{\lambda}\right),$$

where $T$ is the total number of steps.

**Remark 6.2.** Theorem 6.1 gives an $\widetilde{O}(\sqrt{T})$ regret upper bound with respect to the total number of steps $T$. However, for SSP problems, the horizon of each episode is unknown and $T$ can be far greater than $K$. Thus, Theorem 6.1 is not satisfactory due to its dependence on $T$. To deal with this problem, we further prove Theorem 5.2, which translates the dependence on $T$ into the dependence on $K$ but has a worse dependence on the dimension $d$ and other logarithmic factors.

Theorem 6.1 applies to the general cost function with $\rho$ set to 0. Note that the regret upper bound depends on the total number of time steps $T$, which is random. To replace the $T$-dependence by the $K$-dependence, it suffices to show that $T = \widetilde{\mathcal{O}}(K)$. As mentioned in Section 5.1, this can be easily derived under Assumption 5.1. We are now ready to prove Theorem 5.2.

*Proof of Theorem 5.2.* The total cost in $K$ episodes is upper bound by $R_K + KB_\star$ and is lower bounded by $T \cdot c_{\min}$. Together with Theorem 6.1, with probability at least $1 - \delta$, we have

$$T \cdot c_{\min} \leq 6\beta_T\sqrt{dT\log\left(1 + \frac{TB_\star^2}{\lambda}\right)} + 7dB_\star\log\left(T + \frac{T^2B_\star^2d}{\lambda}\right) + KB_\star.$$

Solving the above inequality for the total number of steps $T$, we obtain that

$$T = \mathcal{O}\left(\log^2\left(\frac{1}{\delta}\right) \cdot \left(\frac{KB_\star}{c_{\min}} + \frac{B^2d^2}{c_{\min}^2}\right)\right).$$

Plugging this into Theorem 6.1 yields the desired result. $\qquad\square$

Note that for the general cost functions, by simply picking $\rho = K^{-1/3}$ the result immediately follows from the case of positive costs, which is summarized in Theorem 5.4.

## 6.2 PROOF SKETCH OF THEOREM 6.1

The main steps in proving Theorem 6.1 include an analysis of `EVI` and a regret decomposition. The complete proof can be found in Appendix D.

**Analysis of `EVI`.** By the algorithmic design we elaborated in Section 4, `EVI` guarantees optimism and finite-time convergence, which is summarized in Lemma 6.3 below.

**Lemma 6.3.** Let $\rho = 0$ and $\beta_t = B\sqrt{d\log\left(4(t^2 + t^3B^2/\lambda)/\delta\right)} + \sqrt{\lambda d}$ for all $t \geq 1$, where $B \geq B_\star$. Then with probability at least $1 - \delta/2$, for all $j \geq 1$, `EVI` converges in finite time and the following holds

$$\boldsymbol{\theta}^* \in \mathcal{C}_j \cap \mathcal{B}\,, \quad 0 \leq Q_j(\cdot, \cdot) \leq Q^\star(\cdot, \cdot)\,, \quad \text{and} \quad 0 \leq V_j(\cdot) \leq V^\star(\cdot)\,.$$

Note that in Lemma 6.3 the optimism only holds for the `EVI` output, i.e., $V_j$ for any $j \geq 1$. The initialization $V_0$ in Line 2 of the main Algorithm 1 does not necessarily satisfy the optimism since it is possible that $V^\star(s) < 1$ for some $s$. Still, such an initialization guarantees $\|V_0\|_\infty = 1 \leq B_\star$, which is crucial for establishing the optimism for $j \geq 1$. The proof of Lemma 6.3 is given in Appendix F.1.

**Regret Decomposition.** In our analysis, instead of dealing with (3.3) directly, we first implicitly decompose the times steps into intervals, which are indexed by $m = 1, \ldots, M$ in Lemma 6.4 below. The basic idea here is to decompose all the time steps into disjoint intervals of which the end points are either the end of an episode or the time steps when the `EVI` subroutine is triggered [3]. The purpose of such a regret decomposition is to guarantee that within each interval the optimistic action-value function remains the same, so the induced policy. This is a necessary and common requirement and can be found in the case of discounted infinite horizon MDPs (Zhou et al., 2021b). Similar decomposition trick has also been used in existing works on SSP (Rosenberg et al., 2020; Rosenberg & Mansour, 2020; Tarbouriech et al., 2021b).

**Lemma 6.4.** Assume the event in Lemma 6.3 holds, then we have the following upper bound for the regret defined in (3.3) [4]:

$$R(M) \leq \underbrace{\sum_{m=1}^{M}\sum_{h=1}^{H_m} \left[c_{m,h} + \mathbb{P}V_{j(m)}(s_{m,h}, a_{m,h}) - V_{j(m)}(s_{m,h})\right]}_{E_1}$$

$$+ \underbrace{\sum_{m=1}^{M}\sum_{h=1}^{H_m} \left[V_{j(m)}(s_{m,h+1}) - \mathbb{P}V_{j(m)}(s_{m,h}, a_{m,h})\right]}_{E_2} \tag{6.1}$$

$$+ 2dB_\star \log\left(1 + \frac{TB_\star^2 d}{\lambda}\right) + 2B_\star \log(T) + 2.$$

**Bounding $E_1$ and $E_2$** We bound the terms $E_1$ and $E_2$ separately. Note that $E_2$ is the sum of a martingale difference sequence, and can be bounded by $\mathcal{O}(\sqrt{T\log(T/\delta)})$ using standard concentration. Bounding $E_1$ is more technical and it requires almost all the properties of our algorithmic design. In detail, we need to show that every time when `EVI` is triggered, it can output an optimistic action-value function estimator with high probability (by Lemma 6.3). Second, we need to bound the total difference between the estimated functions and the optimal action-value function. This follows from the elliptical potential lemma and the determinant-based doubling criterion. Third, we need to bound the length of the epochs (i.e., the number of time steps between two `EVI`s), which is achieved by the time-step doubling criterion as explained in Section 4.

## 7 CONCLUSIONS

In this paper, we proposed a novel algorithm for linear mixture SSP and proved its regret upper and lower bounds. For future work, there are several important directions. First, there is a $B_\star^{0.5}$ gap between the current upper and lower bounds. We believe this gap can be closed by using a Bernstein-type of confidence set (Zhou et al., 2021a). Second, it remains open to prove a $\widetilde{\mathcal{O}}(\sqrt{K})$ regret bound for linear mixture SSPs for general cost functions when $c_{\min} = 0$.

---

[3] The interval decomposition is indexed by $m$ in Lemma 6.4. It is implicit and only for the purpose of analysis. This is different from the epoch decomposition, which is explicit and indexed by $j$ in Algorithm 1. The difference is that an epoch ends when `EVI` is triggered, while an interval ends when either `EVI` is triggered or the goal state $g$ is reached (i.e., an episode ends).

[4] $R(M)$ is the same as $R_K$. We use a different notation to emphasize the interval decomposition.

## ETHICS STATEMENT

We don't see any potential ethical issues in our work.

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
