# OpenReview forum: "Learning Stochastic Shortest Path with Linear Function Approximation"
_ICLR.cc/2022/Conference — ICLR 2022 Submitted_

### Official Review · Reviewer_q9Tm · 2021-10-28

**Correctness:** 4
**Technical Novelty And Significance:** 3
**Empirical Novelty And Significance:** 1
**Recommendation:** 6
**Confidence:** 3

**Main Review:**

This work is a nice extension of ideas from RL with linear function approximation to the SSP problem and provides a new setting + new algorithm with upper bound on regret and almost matching lower bounds in some settings.

Detailed Comments:
1. Algorithm:
    1. The algorithm, LEVIS, is very simple, easy to follow and clean.
    2. I especially liked the exposition in Section 4, which goes over the subtle details in the algorithm and places the contribution of the paper in relation to known works.

2. Proof Sketch
    1. In my opinion, Proof of Theorem 5.2 is not insightful and probably should be moved to Appendix.
    2. Instead, I think it will be useful to add more technical detail to paragraph about ``Bounding $E_1$ and $E_2$'' and defining the notation for this discussion. Especially, it will be useful to place the technical contribution of this paper in relation to known techniques like elliptical potential lemma (in more technical detail than Section 4).

3. Minor Issues:
    1. T is only explicitly defined in Section 5.1 but is used in Section 4.
    2. Mapping j() is only defined in the appendix but is used in Equation 6.1.
    3. In proof of lemma C.1, it will be useful to define $P_{m,h}$ under equation C.1, instead of defining it after its used later. But overall the proofs are very detailed.

4. Verified Proofs:
I tried to verify all the proofs. For some proofs, I only skimmed through the proof and followed the general steps, without verifying each step. Note that even when I state that I have verified a proof, it is still possible that I did not catch a subtle issue.

    1. Lemma 6.3: Verified.
    2. Lemma B.1: Verified.
    3. Lemma B.2: Verified.
    4. Lemma C.1: Verified.
    5. Lemma C.2: Verified
    6. Theorem 5.4: Skimmed at high level without verifying all steps.

**Summary Of The Paper:**

This paper studies the linear mixture SSP problem, where the transition kernel is a mixture of unknown models. This paper builds on ideas from recent results in RL with linear function approximation and uses these to introduce a new setting: linear mixture SSP problem + new algorithm, LEVIS for this setting.

Algorithm:
LEVIS is similar to UCB/UCLK style algorithms, however instead of updating the policy every time, the updating is delayed/lazy until either the volume of the feature set (measured by the determinant) grows by a constant factor or a constant multiple of time passes. To update the policy, a subroutine EVI is used which is a slight variation of the optimistic planning. Here the optimistic Q function is multiplied by a $1 - 1/t < 1$ factor to ensure the contraction property while only adding a small additive bias.

This work has three contributions:
1. New Setting and Algorithm: Introduces a new setting for SSP problem: linear mixture SSP problem and a novel algorithm LEVIS for this setting.
2. Upper bound: Under assumption that cost is lower bounded by $c_{\textrm{min}}>0$, LEVIS achieves $O(B_*^{1.5} d \sqrt{K/c_{\textrm{min}}})$ regret. When $c_{\textrm{min}}=0$, a weaker bound is provided $O(K^{2/3})$.
3. Lower bound: Any algorithm for linear mixture SSP problem must have a regret at least $\Omega(d B_* \sqrt{K})$. This is optimal in factors of $d$ and $K$ when $c_{\textrm{min}}>0$.

**Summary Of The Review:**

I think overall this is a good paper building on ideas from RL with linear function approximation to SSP setting. There are subtle issues with extending these ideas to SSP which this paper resolves.

---

> ### Author Response · Authors · 2021-11-17
> **Response to Reviewer q9Tm**
>
> We would like to thank the reviewer for the effort in verifying our proofs in detail.
>
> ---
>
> ***Q1***. T is only explicitly defined in Section 5.1 but is used in Section 4.
>
> ***A1***. Thanks for pointing this out. We will define it the first time it appears in the revision.
>
> ---
>
> ***Q2***. Mapping $j()$ is only defined in the appendix but is used in Equation 6.1.
>
> ***A2***. We have actually introduced the notion $j(\cdot)$ in Section 4. In the revision, we will recap the definition of $j(\cdot)$ before Eq. 6.1.
>
> ---
>
> ***Q3***. In proof of lemma C.1, it will be useful to define $P_{m,h}$ under equation C.1, instead of defining it after it is used later.
>
> ***A3***. Thank you for the suggestion. We will change it accordingly.
>
> We have fixed all the above issues in the revision.

---

### Official Review · Reviewer_sn7Z · 2021-11-02

**Correctness:** 4
**Technical Novelty And Significance:** 2
**Empirical Novelty And Significance:** Not applicable
**Recommendation:** 6
**Confidence:** 3

**Main Review:**

Strengths:

The paper is overall well written. The problem formulation and Assumptions are clearly presented. It is in general easy to follow the paper.

There is an extensive literature review and references to many related works on SSP and mixture models.

The formulation of the SSP problem under linear mixture models seems novel.


Weaknesses:

Although the paper is well written in general, there are places where the writing can improve:
- the rationale behind some steps borrowed from the existing work is not clear. I think some explanations can help the paper to be self-contained. For example the use of discount factor q in the bottom of page 6,  perturbation trick, the criteria of doubling determinant
-  The term action-value function is used for Q function, but later it is changed to Q function

The theoretical results (which is the main contribution of the paper as there are no experiments), although new, seem to be a natural mix of existing ideas. In fact the confidence intervals used for the linear mixture models seem to be the classic confidence ellipsoids first introduced in the context of linear bandits, which have been used in various linear bandit and RL problems. Overall, the algorithmic and analytical contributions seem to be adapting the results of (Zhou et al., 2021a;b) to the SSP setting. I am not certain if this constitutes a major contribution. I appreciate it if authors can provide more details on the challenges of their analysis beyond the existing work.

I am not familiar with the motivation behind assuming a linear mixture model. It seems to make more sense to mix random variables rather than distributions. As mentioned by authors, the mixture may not even be a distribution. Can authors comment on the motivation of this model?

It seems that the bounds can still significantly improve.

- Assuming a $c_{min}>0$ allows the assumption of $B=O(B^*)$. That trivially removes the dependence on the length of the episode. Please let me know if this is correct.
- As mentioned by the authors, standard Bernstein-type variation of confidence intervals can be used to further improve the regret bounds.






**Summary Of The Paper:**

This paper considers the stochastic shortest path (SSP) problem, a particlar RL problem, where a cost function $c(s,a)$ is associated with a state-action pair. A cost-to-go (value function) is defined as the total expected cost of reaching a goal state from an initial state. Then an episodic regret is used to measure the performance. This framework seems to have been considered in several existing works including Cohen et al. (2021) and Tarbouriech et al. (2020b), in a tabular setting. This paper considers the SSP problem under a a "linear
mixture" model of the transition kernel. A model based algorithm is introduced using optimistic estimators of optimal Q function where the stimated Q function is updated through a subroutine called Extended Value Interation (EVI). Upper bounds and a lower bound on regret are proven.

**Summary Of The Review:**

The authors consider SSP frameork under the linear mixture models (that is particular structure for the transition kernel). Adapting ideas from recent works on linear mixture modes in other settings to SSP setting, a model based algorithm is presented and its regret bounds are provided.

---

> ### Author Response · Authors · 2021-11-17
> **Response to Reviewer sn7Z (part 2)**
>
> ***Q4***. Can the current result be improved? For example, assuming a $c_{min}>0$, allows the assumption of $B=\mathcal{O}(B^{*})$. That trivially removes the dependence on the length of the episode. Please let me know if this is correct.
>
> ***A4***. We think the reviewer’s question is: if we assume a $c_{min} > 0$ and $B > B^{*}$, whether we can get a trivial upper bound on the length of each episode (e.g., $B/c_{min}$) and thus remove the dependence on such length.
>
> If our understanding is correct, then the answer is actually negative. This is because by definition $B^{*}$ is the expected cost of the optimal policy for the SSP instance. With $c_{min}>0$ one can only bound the *expected* total time steps of the optimal policy. But in reality, the length of an episode can still be infinite. This is a unique feature of the SSP problem that is different from finite-horizon MDPs.
>
> ---
>
> ***Q5*** As mentioned by the authors, standard Bernstein-type variation of confidence intervals can be used to further improve the regret bounds.
>
>
> ***A5***
> We think it is possible to design a Bernstein-type algorithm to further improve the dependence on $B$ from $\mathcal{O}(B^{1.5})$ to $\mathcal{O}(B)$, matching the dependence on $B$ in the lower bound. Our belief is based on two facts. First, this $\mathcal{O}(B)$ dependence can be obtained for the tabular SSP with Bernstein-type algorithms (see, for example, Rosenberg et al., 2020 and Tarbouriech et al., 2021b). Second, as a comparison, in finite horizon linear mixture MDPs, a Bernstein-type algorithm proposed in Zhou et al. (2021) achieves $\mathcal{O}(H)$ dependence. The key idea is to construct another linear estimator to estimate the variance of transition probability, and use weighted ridge regression to estimate the model parameter. Since $B^{*}$ in SSP can be viewed as a counterpart of $H$ in finite horizon MDPs, we think a similar result is achievable in SSP by extending our algorithm in a way similar to that in Zhou et al. (2021).
>
> For the ease of presentation, we consider the Hoeffding-type algorithm and its analysis, which we believe already conveys the central idea and novelty of our approach. We will study the Bernstein-type algorithm in our future work. We have added a discussion in Appendix A.2 about the extension to the Bernstein-type algorithm.

---

> ### Author Response · Authors · 2021-11-17
> **Response to Reviewer sn7Z (part 1)**
>
> Thank you for the positive comments and suggestions!
>
> ---
>
> ***Q1*** The theoretical results, although new, seem to be a natural mix of existing ideas.
>
> ***A1***
> We think the reviewer might overlook the novelty of our work. Our algorithmic design and analysis are novel in the following aspects.
>
> First, our EVI algorithm (Algorithm 2) is different from that for the discounted linear mixture MDPs. For the discounted setting, the convergence of EVI is guaranteed by the fixed discounted factor $0<\gamma < 1$, which makes the Bellman operator a contraction. However, in linear mixture SSP we no longer have such a property. To deal with this issue, we add a time-varying auxiliary discount factor i.e., $q$ in Eq. (4.1), which is carefully chosen to ensure a balance between convergence rate and bias.
>
> Second, our updating condition (Line 9 of Algorithm 1) is novel, which consists of two criteria: (1) the time-step doubling criterion and (2) the determinant doubling criterion. As explained in Section 4, the former ensures the finite length of each epoch, which cannot be guaranteed by the determinant doubling criterion alone. Note that bounding the length of each epoch is not an issue for finite-horizon MDPs or discounted MDPs. Also, for the tabular SSPs, the visitation doubling condition suffices due to the finite cardinality of the state-action space.
>
> As can be seen from the above, we have proposed several novel algorithm designs that are new and tailored to the (linear) SSP problems. These new algorithm designs also require new analyses in different places of the proof.
>
> For example, in Lemma 6.2 we need to prove the convergence and optimism of our EVI simultaneously instead of separately. This is different from the proof of the discounted MDPs, and such difference is caused by the fact that the value function estimators in the EVI for discounted MDPs are upper bounded by $1/(1-\gamma)$ with probability 1, while unfortunately the value function estimators is our EVI is not necessarily always upper bound by $B^{*}$. This unboundedness prevents us from directly applying the lemma for self-normalized vector-valued martingales (i.e. Lemma E.1). Therefore in the proof of Lemma 6.2 (see Appendix E), we carefully use an induction argument to overcome this challenge.
>
> For another example, in the above we mentioned that we use the time-step doubling criterion to bound the length of each epoch, since one cannot estimate how long it takes the determinant to double. However, we also hope this extra criterion cannot increase the total number of calls of EVI by too much. In the proof of Lemma B.1 we see this is indeed satisfied by the time-step doubling criterion. As a comparison, in tabular SSP one uses the visitation doubling criterion and keeps track of the visitation of all state-action pairs (see for e.g. the proof of Lemma 31 in Tarbouriech et al., 2021b).
>
> ---
>
> ***Q2***. Add explanation for the following: the rationale behind some steps borrowed from the existing work is not clear. I think some explanations can help the paper to be self-contained. For example the use of discount factor $q$ in the bottom of page 6, perturbation trick, the criteria of doubling determinant.
>
> ***A2***. Thank you for your suggestion! We have added more explanations in the revision.
>
> ---
>
> ***Q3***. What are the motivations behind linear mixture models? Any examples?
>
> ***A3***. The linear mixture model is a commonly considered model for linear function approximation. The linear mixture model is both intuitive and covers several important MDP models studied in the literature. We briefly discuss them here.
> First of all, consider an example where the unknown transition model $\mathbb{P}$ belongs to a family of base models $\{ \mathbb{P}_{j} \}$, ${j \in [d]}$ which is available to the learning agent.
>
> Then the true model can be written as a linear combination of models in the family as $\mathbb{P} = \sum_{j \in d}\theta_j \mathbb{P}_j $. In this case, the linear mixture model’s formulation can be viewed as a way of aggregating a large family of known basis models (Modi et al, 2019).
>
> Second, the linear mixture model also covers other important MDP models including the most basic tabular MDP model, as well as the linear-factored MDP model by Yang and Wang, (2020).
>
> For more details, one can look at, for example, Section 2 in Ayoub et al., 2020, or Section 3 in Zhou et al, 2021b.
>
> We have added a discussion in Appendix A.1 about linear mixture models.

---

### Official Review · Reviewer_wXVK · 2021-11-02

**Correctness:** 4
**Technical Novelty And Significance:** 2
**Empirical Novelty And Significance:** Not applicable
**Recommendation:** 5
**Confidence:** 3

**Details Of Ethics Concerns:**

Does not apply!

**Main Review:**


**Strengths.**

  - The problem of SSP with linear function approximation is a natural extension of the SSP problem in the tabular case and there is not much work on this. Therefore, I think it is interesting to study such a problem.
  - The overall presentation of this paper is great and the theoretical results appear to be sound and clear as far as I have checked.
 - The authors provide also a minimax lower bound which is very much appreciated.

**Weaknesses.**

  -  The analysis and technical tools used for sequential decision-making problems under the linear structure assumption are by now becoming quite standard. The current work applies such analysis and tools to the SSP problem in an immediate manner. Therefore, this work lacks in terms of originality and novelty. The authors discuss the use of Bernstein-type inequality for the tabular case. It would be a great addition if a bound similar in spirit could be derived for the linear case. Furthermore, perhaps the authors could also highlight the technical challenges arising in this setting.
 - The provided regret bounds appear to be loose than what one would expect.For example:  (i) the regret bound has a dependency of order $\log^2(KBd^2/\delta)$, while one would expect a dependency of order $\sqrt{\log(KBd^2/\delta)}$ (as in linear bandits for instance); (ii) The authors did not discuss the low order dependencies appearing in the additive term in the regret bound $ \frac{B^2d^2}{c_\min} \log2(KBd^2/\delta)$. I would be curious to know whether $d^2$ could be brought down to $d$ and whether the dependency on $c_{\min}$ could be avoided?
  - The proposed algorithm requires as input an upper bound $B$ on the unknown $B_\star$, and a regularisation parameter $\lambda$, and cost perturbation $\rho$. The authors do not discuss how these parameters should be chosen. More specifically, I am curious to know how should $B$ be chosen?
  - The authors provide an algorithm with a regret upper bound in high probability while the provided lower bound is for the expected regret. This is a standard practice in the literature but it remains an unsatisfactory presentation since one would hope to get a lower bound for regret in high probability. Perhaps the author could comment on this.

**Summary Of The Paper:**

This work considers the stochastic shortest path (SSP) problem with linear function approximation. The authors propose an algorithm which they refer to as LEVIS, and provide a regret guarantee of order $R_K = O\left(\left(B^{1.5}d \sqrt{K/c_{\min}}  + \frac{B^2d^2}{c_{\min}} \right) \log^2(KBd/c_{\min}\delta) \right)$. The algorihtm relies on the principle of optimism in face of uncertainty. Finally, the authors also provide a minimax lower bound on the expected regret of order $\Omega(dB_\star \sqrt{K})$.

**Summary Of The Review:**

My recommendation of the paper is based on the following (more details in the main review):
- (+) I appreciate such a study because there is a lack of existing work on SSP with linear function approximation as far as I can tell (except the work of Vial et al. 2021).
- (-) My first concern is that I do not see any novelty or significant contribution on the technical front. Most of the technical tools and ideas used are borrowed from prior work with minor changes.
- (-) My second concern is that the obtained guarantee appears to have some loose dependencies on the problem parameters and confidence parameter $\delta$. A discussion on such dependencies is lacking, especially in the low-order terms appearing in the regret upper bound (Theorem 1).
- (-) My third concern is that the proposed algorithm requires certain inputs which might rely on unknown quantities. The authors did not clarify how these inputs should be chosen.

---

> ### Author Response · Authors · 2021-11-17
> **Response to Reviewer wXVK (part 2)**
>
> ***Q3***. The provided regret bounds appear to be loose than what one would expect. For example, (i) the regret bound has a dependency of order  $\log^2⁡(KB d^2/\delta)$, while one would expect a dependency of order  $\sqrt{\log ⁡(KB d^2/\delta)}$ (as in linear bandits for instance)
>
> ***A3***.
> First of all, we would like to clarify that even in linear bandits, the dependence should be $\log ⁡(\cdots)$ rather than $\sqrt{\log ⁡(\cdots)}$ dependence. (See for example Theorem 3 in Abbasi-Yadkori et al., 2011. The dependence is essentially $\log(\cdots)$.)
>
> Second, to compare our results with those for linear bandits, it’s clearer to use Theorem 6.1 instead of Theorem 5.2. Indeed, in Theorem 6.1, the dependence is $\log(KB d^2/\delta)$ instead of $\log^2 (KB d^2/\delta)$, which aligns with the dependence in linear bandits. Note that if the horizon for each episode is ‘known’ a priori, then the bound of Theorem 6.1 matches the intuition from linear bandits.
>
> However, for SSP problems, the horizon of each episode is unknown and thus Theorem 6.1 is not good enough due to its dependence on T, which is unknown and random. Therefore, in order to translate it into dependence on $K$, we need to solve an inequality as explained in the proof of Theorem 5.2. This $\log^2 (\cdots)$ dependence inevitably occurs when we solve this inequality.
>
> It is also worth mentioning that in finite horizon MDPs, one has $T = KH$. We can then avoid the extra $\log(\cdots)$ factor by simply plugging $T = KH$ into the bound of Theorem 6.1, but unfortunately, SSP does not have such a property.
>
> Therefore, we view this $\log(\cdots)$ dependence as a natural consequence of the unknown horizon length of the SSP problem.
>
> ---
> ***Q4***.  (ii) The authors did not discuss the low order dependencies appearing in the additive term in the regret bound $B^2 d^2 c_{min} \log^⁡2 (KB d^2/ \delta)$. I am curious to know whether $d^2$ could be brought down to $d$ and whether the dependency on $c_{min}$ could be avoided?
>
> ***A4***.
> For the dependence on $d$ being $d^2$ instead of $d$, this is due to the same reason as in ***A3*** (note that in Theorem 6.1, it is also $d$ instead of $d^2$).
> The dependence on $c_{min}$ cannot be removed either since it also comes from solving the inequality.
>
> ---
> ***Q5***. The proposed algorithm requires as input an upper bound $B$ on the unknown $B^{*}$, and a regularisation parameter $\lambda$, and cost perturbation $\rho$. The authors do not discuss how these parameters should be chosen. More specifically, I am curious to know how $B$ should be chosen?
>
> ***A5***. Thank you for pointing this out!
>
> For $\lambda$, by Theorems 5.2 and 5.3, one can always pick $\lambda = 1$.
>
> For $\rho$, there are two cases. When $c_{min}>0$, one can simply pick $\rho = 0$. When $c_{min}=0$, by Theorem 5.3, one can pick $\rho = K^{-1/3}$.
>
> As for $B$, we would like to mention that the assumption of a known upper bound
> $B \geq B^*$ or even a known $B^{*}$ is common in existing SSP literature. For example, this assumption is made in the linear SSP studied by Vial et al., (2021).
> Although it is possible to deal with the issue of unknown $B$ in the tabular SSP with the doubling trick (for example in Rosenberg et al., 2020 and Tarbouriech et al., 2021b), it remains an open question for SSP with linear function approximation.
>
> We have added more discussions on the choice of $\lambda, \rho$, and $B$ in the revision.
>
> ---
> ***Q6***. The authors provide an algorithm with a regret upper bound in high probability while the provided lower bound is for the expected regret. This is a standard practice in the literature but it remains an unsatisfactory presentation since one would hope to get a lower bound for regret in high probability. Perhaps the author could comment on this.
>
> ***A6***.
> The difference between the high probability regret upper bound and the expected regret lower bound is not an issue. For the upper bound, if we choose the probability $\delta=1/K$, then the high probability upper bound can imply an $\tilde{O}(d\sqrt{K})$ expected regret bound. This result matches the lower bound in terms of the dependence on $d$ and $K$, and it suggests that the lower bound can not be improved in $d$ and $K$.
>
> Similarly, if there exist an algorithm that can obtain a smaller regret than $\tilde{\Omega}(d\sqrt{K}\log(1/\delta))$ with probability at least $1-\delta$ for any $0<\delta<1$, then it implies that the expected regret of this algorithm is smaller than $\tilde{\Omega}(d\sqrt{K})$, which contradicts to our lower bound. Therefore, our lower bound shows that the upper bound can not be improved in its dependence on $d$ and $K$.
>
> Finally, while there exists no high probability regret lower bound even for linear bandit problem, we think such a lower bound can be proved by following the technique used in Chapter 17 in Bandit Algorithms (Lattimore and Szepesvari 2020). We will leave it in our future work.

---

> ### Author Response · Authors · 2021-11-17
> **Response to Reviewer wXVK (part 1)**
>
> Thank you for the detailed comments and suggestions!
>
> ---
>
> ***Q1***. What is the technical novelty?
>
> ***A1***. We think the reviewer might overlook the novelty of our work. Our algorithmic design and analysis are novel in the following aspects.
>
> First, our EVI algorithm (Algorithm 2) is different from that for the discounted linear mixture MDPs. For the discounted setting, the convergence of EVI is guaranteed by the fixed discounted factor $0<\gamma < 1$, which makes the Bellman operator a contraction. However, in linear mixture SSP we no longer have such a property. To deal with this issue, we add a time-varying auxiliary discount factor i.e., $q$ in Eq. (4.1), which is carefully chosen to ensure a balance between convergence rate and bias.
>
> Second, our updating condition (Line 9 of Algorithm 1) is novel, which consists of two criteria: (1) the time-step doubling criterion and (2) the determinant doubling criterion. As explained in Section 4, the former ensures the finite length of each epoch, which cannot be guaranteed by the determinant doubling criterion alone. Note that bounding the length of each epoch is not an issue for finite-horizon MDPs or discounted MDPs. Also, for the tabular SSPs, the visitation doubling condition suffices due to the finite cardinality of the state-action space.
>
> As can be seen from the above, we have proposed several novel algorithm designs that are new and tailored to the (linear) SSP problems. These new algorithm designs also require new analyses in different places of the proof.
>
> For example, in Lemma 6.2 we need to prove the convergence and optimism of our EVI simultaneously instead of separately. This is different from the proof of the discounted MDPs, and such difference is caused by the fact that the value function estimators in the EVI for discounted MDPs are upper bounded by $1/(1-\gamma)$ with probability 1, while the value function estimators in our EVI are not necessarily always upper bounded by $B^{*}$. This unboundedness prevents us from directly applying the lemma for self-normalized vector-valued martingales (i.e. Lemma E.1). Therefore in the proof of Lemma 6.2 (see Appendix E), we carefully use an induction argument to overcome this challenge.
>
> For another example, in the above, we mentioned that we use the time-step doubling criterion to bound the length of each epoch since one cannot estimate how long it takes the determinant to double. However, we also hope this extra criterion cannot increase the total number of calls of EVI by too much. In the proof of Lemma B.1 we see this is indeed satisfied by the time-step doubling criterion. As a comparison, in tabular SSP one uses the visitation doubling criterion and keeps track of the visitation of all state-action pairs (see for e.g. the proof of Lemma 31 in Tarbouriech et al., 2021b).
>
> ---
>
> ***Q2***. Discuss the use of a Bernstein-type algorithm.
>
> ***A2***.
> We think it is possible to design a Bernstein-type algorithm to further improve the dependence on $B$ from $\mathcal{O}(B^{1.5})$ to $\mathcal{O}(B)$, matching the dependence on $B$ in the lower bound. Our belief is based on two facts. First, this $\mathcal{O}(B)$ dependence can be obtained for the tabular SSP with Bernstein-type algorithms (see, for example, Rosenberg et al., 2020 and Tarbouriech et al., 2021b). Second, as a comparison, in finite horizon linear mixture MDP, a Bernstein-type algorithm proposed in Zhou et al. (2021) achieves $\mathcal{O}(H)$ dependence. The key idea is to construct another linear estimator to estimate the variance of transition probability, and use weighted ridge regression to estimate the model parameter. Since $B^{*}$ in SSP can be viewed as a counterpart of $H$ in finite horizon MDPs, we think a similar result is achievable in SSP by extending our algorithm in a way similar to that in Zhou et al. (2021).
>
> For the ease of presentation, we consider the Hoeffding-type algorithm and its analysis, which we believe already conveys the central idea and novelty of our approach. We will study the Bernstein-type algorithm in our future work. We have added a discussion in Appendix A.2 about the extension to the Bernstein-type algorithm.

---

### Decision · Program_Chairs · 2022-01-20

**Decision:**

Reject

**Comment:**

This paper studies the stochastic shortest path (SSP) problem with a linear approximation to the transition model. The authors propose a doubling algorithm for regret minimization in this setting and bound its regret. This is a theory paper with no experiments.

This paper received three borderline reviews. All reviewers agreed on its strengths and weaknesses during the discussion. The strengths are that the paper is well written and that the results are novel. The weaknesses are that the proposed solution is standard and analyzed using standard tools. The reviewers noted departures from the standard analyses but these seem to be minor technical issues. Therefore, although well executed, this paper lacks novelty. No reviewer argued for the acceptance of this paper and therefore it is rejected.